# Effects of Space Flight on Mouse Liver versus Kidney: Gene Pathway Analyses

**DOI:** 10.3390/ijms19124106

**Published:** 2018-12-18

**Authors:** Timothy G. Hammond, Patricia L. Allen, Holly H. Birdsall

**Affiliations:** 1Durham VA Medical Center, Medicine Service Line, Durham, NC 27705, USA; plallen08@gmail.com; 2Nephrology Division, Department of Internal Medicine, Duke University School of Medicine, Durham, NC 27705, USA; 3Space Policy Institute, Elliott School of International Affairs, George Washington University, Washington, DC 20052, USA; hhbirdsall@gmail.com; 4Department of Veterans Affairs Office of Research and Development, Washington, DC 20420, USA; 5Departments of Otorhinolaryngology, Immunology, and Psychiatry, Baylor College of Medicine, Houston, TX 77030, USA

**Keywords:** spaceflight, microgravity, gene expression, liver, kidney, gene array, gene expression, transcriptome, apoptosis, Cdkn1a, CYP4A1

## Abstract

Understanding genome wide, tissue-specific, and spaceflight-induced changes in gene expression is critical to develop effective countermeasures. Transcriptome analysis has been performed on diverse tissues harvested from animals flown in space, but not the kidney. We determined the genome wide gene expression using a gene array analysis of kidney and liver tissue from mice flown in space for 12 days versus ground based control animals. By comparing the transcriptome of liver and kidney from animals flown in space versus ground control animals, we tested a unique hypothesis: Are there common gene expression pathways activated in multiple tissue types in response to spaceflight stimuli? Although there were tissue-specific changes, both liver and kidney overexpressed genes in the same four areas: (a) cellular responses to peptides, hormones, and nitrogen/organonitrogen compounds; (b) apoptosis and cell death; (c) fat cell differentiation and (d) negative regulation of protein kinase.

## 1. Introduction

Transcriptome and proteasome analysis has been performed on diverse tissues harvested from several animal species flown in space. Many gene and gene pathway changes associated with changes due to the diverse stimuli of spaceflight have been identified: microgravity, radiation exposure, reduction, and reduction of convection, to say nothing about the effects of launch, maneuvers, and landing. 

By analyzing the transcriptome of liver and kidney from animals flown in space versus ground control animals, we can test a unique hypothesis: Are there tissue specific gene expression pathways activated in individual tissue types in response to spaceflight stimuli? For pathways that are common, do the same genes predominate, or is the distribution of contributing pathway specific genes differ? Using liver and kidney harvested from the same mice after spaceflight and identically housed ground controls, direct comparison can be made between tissues without confounding effects of sex, strain, species, and age, while minimizing differences in microbiome and epigenetics. 

The mice flew on space shuttle flight STS-108, also known as Utilization Flight 1 (UF-1), to the International Space Station. Data were not published immediately due to a laboratory loss occurred during Hurricane Katrina. We present the data now because contemporary analysis methods can be applied to this unique data set to directly address unanswered but relevant questions.

## 2. Results

### 2.1. Gene Arrays

Liver and kidney tissues showed similar gene response to spaceflight. Figure 1 shows responses of 12,489 genes in liver versus kidney, expressed as a fold change in space flown versus ground controls, as analyzed by Affymetrix hybridization, along with representative heat maps for the two tissues. (Fold change, average expression, t scores, *p*-values, and adjusted p values for each gene can be found in Appendix A.) Of note, CYP4A1 was increased in space-flown liver, whereas over 20 other CYP genes did not change.

### 2.2. Gene Set Enrichment Analysis (GSEA)

Gene Set Enrichment Analysis (GSEA) provides a computational method to determine whether an *a priori* defined set of genes shows statistically significant and concordant differences between spaceflight and ground controls. GSEA calculates a score for the enrichment of an entire set of genes rather than a single gene. It does not, however, require setting a cutoff, but does allow for an analysis of the complete data set and thus identifies the set of relevant genes as part of the analysis, providing a robust statistical framework. Using a modified Fisher’s exact test, the replicates are integrated into the analysis so that no data are lost [1]. 

Of the 785 gene ontology (GO) pathways analyzed by GSEA, 764 trended towards concordance between liver and kidney tissue response to spaceflight (Figure 2).

Furthermore, nine pathways were upregulated in the liver and downregulated in the kidney. Five of those pathways were related to T cell activation and the others were related to calcium channel, extracellular matrix structure, heart development, and hormone metabolism (see Table 1 for details). Moreover, 12 pathways were upregulated only in kidney but downregulated in liver, whereas only three of those pathways were related to peptide binding and two were related to coagulation. The others were related to amino acid metabolism, rhodopsin-like receptor, cell division, DNA packaging, Golgi compartment, and cytokinesis (see Table 2 for details). 

Appendix A lists the statistical parameters including enrichment score, negative normalized enrichment score, false discovery rate, and family wise error rate for all 785 GO pathways. 

### 2.3. Cytoscape/ClueGO

ClueGO visualizes the interactions of gene clusters in a functionally grouped network using enrichment maps. Similar to the GSEA analysis, liver and kidney showed striking similarities in the ways that they responded to spaceflight (Figure 3 and Figure 4). Both tissues overexpressed genes in the same four areas: (a) cellular responses to peptides, hormones, and nitrogen/organonitrogen compounds; (b) apoptosis and cell death; (c) fat cell differentiation, and (d) negative regulation of protein kinase. In many cases, the exact same pathways were overexpressed in both tissues, as noted by the clusters identified with arrows in Figure 3 and Figure 4. Additional details of the genes for each node can be found in Appendix A.

## 3. Discussion

By analyzing the transcriptome of liver and kidney from animals flown in space versus ground control animals, we were able to evaluate the extent to which tissues from different organs respond to spaceflight in shared ways versus organ-specific ways. We began by examining gene arrays using Affymetrix hybridization and heat maps. We then analyzed the results through GSEA to look for changes across sets of genes, rather than single genes, in order to identify pathways that were in common between the tissues or unique to the particular organs. Finally, we visualized the data by Cytoscape with ClueGO to identify interactions of gene clusters in kidney versus liver.

The first major finding of this study was that we had not only shown tissue specific gene pathway response to spaceflight, but we have identified those gene pathways.

GSEA analysis of the samples identified significant liver-specific changes in genes for T cell activation, calcium channel, extracellular matrix structure, heart development, and hormone metabolism. It is likely that the upregulated T cell activation genes in the liver came from infiltrating T cells. Activation of T cells in spaceflight has been well-described [2], as has the infiltration of the liver with inflammatory mononuclear cells [3]. GSEA analysis also identified significant kidney-specific changes in peptide binding, amine receptor, amino acid catabolism, rhodopsin-like receptor, coagulation, cell division, DNA packaging, cytokinesis, and Golgi compartment.

The second major finding of this study was that there were common gene pathways upregulated in both the liver and kidney during spaceflight. Both tissues overexpressed genes in the same four areas: (a) cellular responses to peptides, hormones, and nitrogen/organonitrogen compounds; (b) apoptosis and cell death; (c) fat cell differentiation, and (d) negative regulation of protein kinase. Our findings in these areas are consistent with the other reports on the livers from mice flown in space, such as an altered ability to respond to insulin [4], impaired management of reactive oxidation (an integral part of apoptosis) [3,5], dysregulated lipid metabolism [3,4], and altered protein kinases [5].

Blaber et al. performed unbiased, unsupervised, and integrated multi-omic analyses of metabolomic and transcriptomic data sets on liver tissue derived from nine-week old weight-matched female C57BL/6J mice flown in space for 13.5 days [3]. They observed an increased enrichment of genes associated with autophagy and the ubiquitin-proteasome. The current data sets show changes in cell death pathways in both kidney and liver, with an emphasis on apoptosis, but also autophagy. Blaber et al. also reported hepatic negative regulation of protein phosphorylation [3], something we now demonstrate in space flown kidney tissue.

Like Blaber et al., we saw a highly significant increase in Cdkn1a, the cyclin-dependent kinase inhibitor 1A (P21), in hepatic tissue and now report the same observation in kidney tissue. Cdkn1a gene encodes a potent cyclin-dependent kinase inhibitor with a regulatory role in S phase DNA replication, DNA damage repair, and apoptosis. The effect of Cdkn1a activation is apparent in the overexpression of apoptosis-regulating pathways in both liver and kidney tissues in space.

Baba et al. analyzed gene and protein expression of 11 cytochrome P450s and stress-associated molecules in liver tissue from male Sprague–Dawley rats, aged 65 days at launch and flown in space for nine days [6]. The gene and protein expression of stress-related proteins CYP4A1and Cirp gene expression was significantly increased whereas HSP90 and p53 gene expression was significantly decreased in the flight group than in the ground control. Combined with histology, it was concluded that the effects of spaceflight on the liver may be similar to mild cold stress or fasting. From our data, we can confirm a large increase in hepatic CYP4A1 during spaceflight, as 47 other hepatic CYPs did not change in space. This is consistent with the finding of Brown et al. in the rotating wall vessel, that rat hepatocytes maintained CYP functions and albumin production for more than 30 days in suspension culture [7]. In our data, HSP90 and p53 gene expression were decreased in the flight group similar to the observations of Baba et al. [6], but did not reach statistical significance.

Baqai et al. harvested liver, spleen, and thymus from female C57BL/6 mice flown for 13 days in order to study the effects of spaceflight on innate immune function and antioxidant gene expression [5]. During the flight, animals had a reduction in liver, spleen, and thymus masses, compared with ground controls; splenic lymphocyte, monocyte/macrophage, and granulocyte counts were significantly reduced in the flight mice. Baqai et al. concluded that exposure to the spaceflight environment can increase anti-inflammatory mechanisms. No transcriptome analysis was performed. We observed an alpha-beta T-cell activation that is heavily liver-specific, suggesting different immune responses in the liver and kidney during spaceflight.

Martinez et al. found a significant reduction of key gene expression in early T-cell activation in C57BL/6J wild-type female mice of similar age and weight flown in space for 15 days [2]. They demonstrated decreases in gene expression for IL2, IL2Rα, and IFNγ during spaceflight. In our data IL2, IL2Rα, and IFNγ gene expression was decreased in the flight group similar to the observations of Martinez et al., but did not reach statistical significance [2].

Pecaut et al. studied age- and weight-matched, 11-week-old female C57BL/6J mice flown in space for ~13 days [4]. There were increases in liver expression profiles related to fatty acid oxidation with decreases in glycolysis-related profiles. They suggested that, given the clear link between immune function and metabolism in many ground-based diseases, a similar link may be involved in spaceflight-induced decrements. Our data demonstrates changes in lipid metabolism in both liver and kidney tissues in space and further clarifies the gene pathway mediators.

In the kidney, we confirm many of the pathways implicated in responses to spaceflight and rotating wall suspension culture in renal proximal tubular cells [8,9]. During spaceflight, we confirmed changes we previously observed in kidney gene pathways, including apoptosis and cellular response to peptides [8]. During the rotating wall suspension culture of renal proximal tubular cells, we observed changes in phosphorylation and protein kinase activity similar to the changes incurred in spaceflight [9].

Spaceflight is a complex mixture of changes in gravity, convection, radiation exposure, and other stressors. While we do not have a detailed similar date on the isolated stimuli for comparison, changes in cell cycle, DNA packing, and oxygenation are consistent with changes in these pathways observed in a number of other systems [10].

We report tissue-specific gene expression pathways in liver and kidney during spaceflight, and for gene pathways that are common, there is a substantial concurrence of the genes that predominate many pathways in both tissues. The confirmation of data from single tissue findings in the literature suggests that our study is reliable. Utilizing liver and kidney tissue harvested from the same mice after spaceflight and identically housed ground controls, is a powerful protocol to optimally remove confounding effects of sex, strain, species, and age, while minimizing differences in micro biome and epigenetics.

## 4. Materials and Methods

### 4.1. Reagents

All reagents were purchased from Sigma Inc. (St. Louis, MO, USA), unless otherwise noted. 

### 4.2. Spaceflight Conditions and Ground Based Controls.

Working in collaboration with NASA Ames Research Center and Bioserve Space Technologies, female C57BL/6J mice were housed in the animal enclosure module (AEM) with special food bars, redundant lixit water supply, and a waste management system. The experiment consisted of six AEMs each with eight mice per AEM. On the space shuttle, three AEMs flew and three served as ground controls. The protocol was reviewed by the Ames IACUC (that served as the Flight IACUC) and was approved on 8 August 2001. The title of protocol was “Investigation of osteoprotegerin as a treatment for spaceflight induced osteopenia on UF-1:STS-108:CBT M-01 payload” and the protocol number was 01-028A-1.

The mice were in space for 11 days and 19 hours. Animals were removed from the space shuttle, released from the AEMs, euthanized, kidney and liver tissues harvested as rapidly as practical, and snapped frozen in liquid nitrogen. In each group, eight mice were analyzed. The animals were controls in a study conducted by Amgen, after review by NASA Kennedy Space Center’s Institutional Animal Care and Use Committee. The present study examines used tissues discarded from Amgen’s studies and therefore did not undergo separate review by the committee. 

### 4.3. RNA Extraction

Samples were transferred to pre-chilled tubes that contained 10 microliters of RNAseOUT (400 U) (Life Technologies, Carlsbad, CA). Added to the tubes and tissues was 3 mL of Trizol homogenized on ice. RNA was extracted using our standard protocol [8,11]. RNA purity was confirmed by the ratio of OD_260 nm_ to OD _280 nm_, which averaged 1.7 and 1.8, and by 1% agarose gel electrophoresis with 18S and 28S ribosomal bands corresponding to sizes on a ladder.

### 4.4. Gene Arrays

Gene arrays were performed on Affymetrix (Santa Clara, CA, USA) GeneChip® Murine Genome U95Av2.1 (MG-U95Av2.1) set arrays. This array contains transcript coverage of the mouse genome to study the expression level of more than 60,000 murine genes and expressed sequence tags, including ~12,000 sequences characterized in terms of function or disease association. Hybridization controls included *bioB*, *bioC*, *bioD*, and *cre*. Poly A controls were *dap*, *lys*, *phe*, *thr*, and *trp*. Housekeeping/control genes were glyceraldehyde 3-phosphate dehydrogenase, beta-actin, transferrin receptor, and interferon-stimulated gene factor-3. Gene arrays were performed at the Medical College of Georgia.

### 4.5. Data Analysis and Statistics

Affymetrix Gene Chip microarray data underwent strict quality control processing using the affy package [12] in Bioconductor [1] in the R statistical programming environment. Log-scale Robust Multiarray Analysis was used for normalization to eliminate systematic differences across the arrays. Significant differentially expressed genes were identified using a linear regression model with an empirical Bayes method for parameter estimation from the limma package [13] in the Bioconductor. The False Discovery Rate method was employed to control for multiple hypothesis testing. 

To perform GSEA, we used Gene Ontology (GO) pathway gene sets found at the Molecular Signatures Database. The DNA microarray data was compared between space flown and ground control tissues and analyzed as to whether the majority of genes in a set fell in the extremes of the list: the top (overexpressed) and bottom (underexpressed) of the list, corresponding to the largest differences in expression between the two conditions. If the gene set was significantly overexpressed or underexpressed, it was taken to be related to phenotypic differences.

### 4.6. Gene Ontology (GO) Enrichment Analysis

We obtained gene ontology annotations of murine genes using ClueGO. ClueGO is a Cytoscape plug-in that visualizes the non-redundant biological terms for large clusters of genes in a functionally grouped network. ClueGO network is created with kappa statistics and reflects the relationships between the terms based on the similarity of their associated genes. 

GO biological processes that were too specific (containing less than five genes) or too general (containing greater than 300 genes) were excluded from the analysis.

Given a query set of genes, we used the hyper geometric test to obtain a *P* value, estimating the significance with which the set is enriched with genes annotated to a given biological process, relative to a gene universe defined as the set of genes with usable data for both static and rotating conditions. 

### 4.7. Cytoscape Analysis Using ClueGO

We visualized GO enrichment results with enrichment maps that were generated using an approach similar to the Enrichment Map Cytoscape Plugin v1.1 [10,14,15]. In contrast to the plugin, the nodes in each map were clustered with Markov cluster (inflation = 2), using the overlap coefficient computed by the plugin as the similarity metric (coefficients less than 0.5 were set to zero). Nodes in the same cluster were assigned the same node color and a cluster label was determined based on common themes in the processes within the cluster. 

## 5. Conclusions

Liver and kidney show both tissue-specific and shared tissue-common responses to the stress of spaceflight. Liver and kidney overexpressed genes in the same four areas: (a) cellular responses to peptides, hormones, and nitrogen/organonitrogen compounds; (b) apoptosis and cell death; (c) fat cell differentiation and (d) negative regulation of protein kinase. CYP4A1 was increased in space-flown liver, whereas over 20 other CYP genes did not change. Cdkn1a, which encodes a potent cyclin-dependent kinase inhibitor with a regulatory role in S phase DNA replication, DNA damage repair, and apoptosis was particularly significantly overexpressed in both liver and kidney tissues in space.

## Figures and Tables

**Figure 1 ijms-19-04106-f001:**
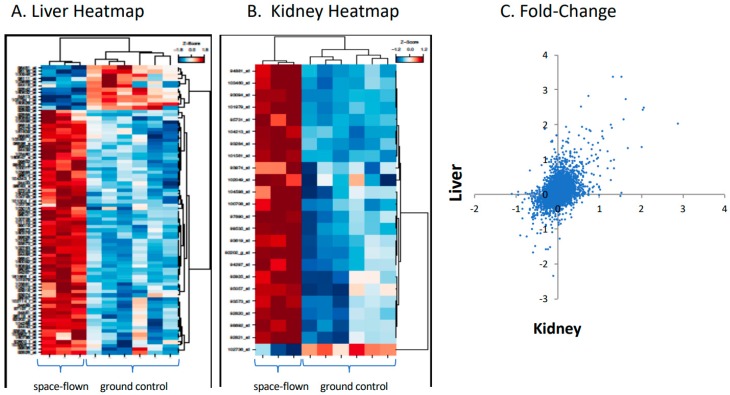
Gene expression for mouse liver and kidney flown in space versus ground controls. Responses of 12,489 genes in mouse liver and kidney, flown in space versus ground controls, as analyzed by Affymetrix hybridization. Panel (**A**) is the heat map of representative liver samples. Panel (**B**) is the heat map for representative kidney samples. Panel (**C**) is the fold-change in gene expression for liver (*y*-axis) plotted against kidney (*x*-axis).

**Figure 2 ijms-19-04106-f002:**
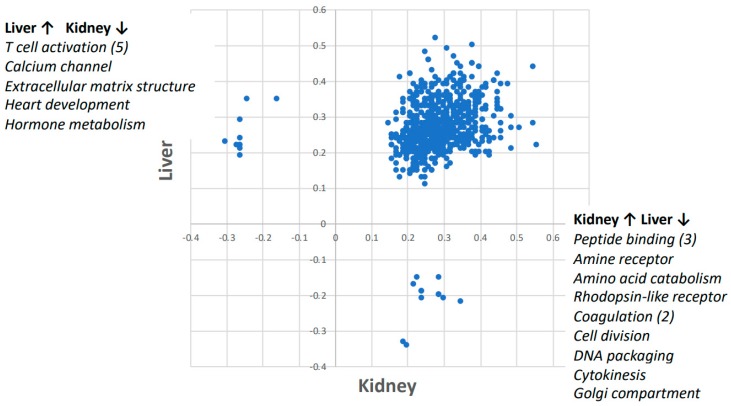
The Gene Set Enrichment Analysis (GSEA) enrichment scores for space-flown vs. ground control mouse liver and kidney. GSEA calculates a score for the enrichment of an entire set of genes across a gene ontology (GO) pathway. A positive score indicates enrichment in the space-flown mice relative to the ground controls. In 764 of the 785 pathways analyzed, there was enrichment in both kidney and liver in response to spaceflight. The nine pathways that were upregulated in liver and downregulated in kidney are listed in the upper left corner and detailed in Table 1. The 12 pathways that were upregulated in kidney but downregulated in liver are listed on the lower right corner and detailed in Table 2.

**Figure 3 ijms-19-04106-f003:**
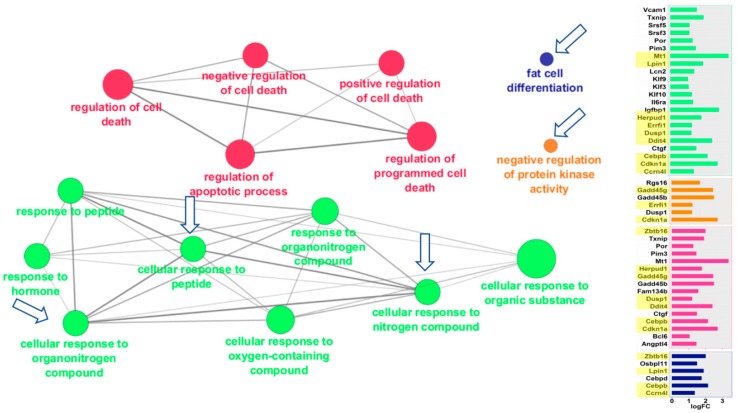
Mouse liver pathways overexpressed in space. Functional gene networks that were upregulated in mouse liver in response to spaceflight. ClueGO Cytoscape visualizes the interactions of gene clusters in a functionally grouped network using enrichment maps. Nodes in the same cluster are assigned the same node color and node size indicates the number of mapped genes in each GO term. The node label is determined on the basis of common themes in the processes within the cluster. Bar graphs of the fold-change (FC) of individuals genes are shown on the right. Gene names highlighted in yellow were also overexpressed in mouse kidney after spaceflight. Arrows mark the gene clusters that were also seen in Space-flown kidney. Additional information on the clusters can be found in Appendix A.

**Figure 4 ijms-19-04106-f004:**
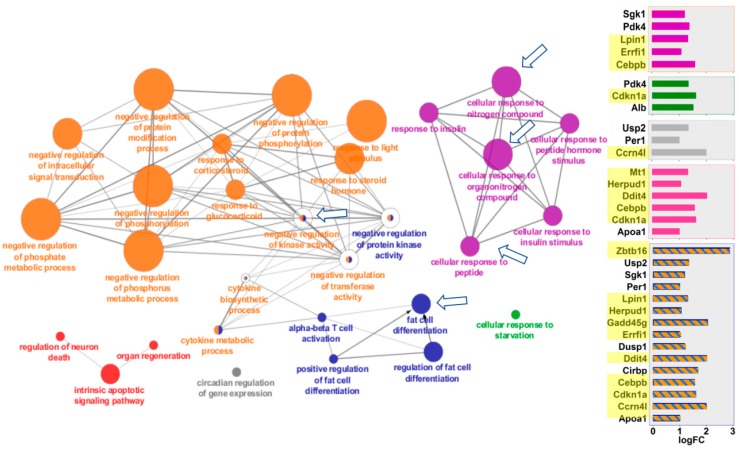
Mouse kidney pathways overexpressed in space. Functional gene networks that were upregulated in kidney in response to spaceflight. ClueGO Cytoscape visualizes the interactions of gene clusters in a functionally grouped network using enrichment maps. Nodes in the same cluster are assigned the same node color and node size indicates the number of mapped genes in each GO term. The node label is determined on the basis of common themes in the processes within the cluster. Bar graphs of the fold-change (FC) of individuals genes are shown on the right. Gene names highlighted in yellow were also overexpressed in mouse liver after spaceflight. Data in the bottom panel with orange and blue stripes applies to both the orange and blue clusters. Arrows mark the gene clusters that were also seen in Space-flown liver. Additional information on the clusters can be found in Appendix A.

**Table 1 ijms-19-04106-t001:** Genes Upregulated in Liver and Downregulated in Kidney during Spaceflight.

Gene Set	Kidney Gene #	Liver Gene #	Kidney ES	Liver ES	Kidney NES	Liver NES	Kidney FWER*p*	Liver FWER*p*	Kidney N*p*	Liver N*p*	Kidney FDR*q*	Liver FDR*q*	Kidney Rank	Liver Rank
T cell activation	29	29	−0.26	0.19	−1.37	0.67	0.338	1	0	0.933	0.186	1	6874	6586
Positive regulation of T cell activation	18	18	−0.26	0.29	−1.09	0.94	0.882	1	0.281	0.559	0.393	0.972	6874	4085
Regulation of Lymphocyte activation	26	26	−0.26	0.21	−1.44	0.75	0.208	1	0	0.869	0.163	1	6874	6695
Positive regulation of Lymphocyte activation	20	20	−0.26	0.22	−1.25	0.74	0.574	1	0.091	0.854	0.19	1	6874	4085
Regulation of T cell activation	21	21	−0.26	0.24	−1.26	0.8	0.535	1	0.071	0.786	0.206	0.997	6874	4085
Calcium channel activity	20	20	−0.27	0.22	−1.36	0.73	0.341	1	0.111	0.857	0.141	1	5840	2858
Extracellular matrix structural constituent	15	15	−0.24	0.35	−1.03	1.09	0.946	1	0.526	0.343	0.453	0.873	6985	4443
Heart development	24	24	−0.3	0.23	−1.49	0.78	0.173	1	0	0.828	0.272	1	6497	4327
Hormone metabolic process	16	16	−0.16	0.35	−0.65	1.1	0.999	1	0.919	0.336	0.97	0.835	7751	4932

Gene # is the number of genes in the pool. ES is the enrichment score. NES is the normalized enrichment score. FWER*p* is the Family Wise Error Rate *p*-value. N*p* is the Normalized *p*-value. FDR*q* is the False Discovery Rate *q*-value. Rank is the Rank at Maximum.

**Table 2 ijms-19-04106-t002:** Genes Upregulated in Kidney and Downregulated in Liver during Spaceflight.

Gene Set	Kidney Gene #	Liver Gene #	Kidney ES	Liver ES	Kidney NES	Liver NES	Kidney FWER*p*	Liver FWER*p*	Kidney N*p*	Liver N*p*	Kidney FDR*q*	Liver FDR*q*	Kidney Rank	Liver Rank
Amine receptor activity	23	23	0.24	−0.21	0.79	−1.01	1	0.983	0.8	0.55	1	0.964	2860	7304
Amino acid catabolic process	19	19	0.22	−0.17	0.68	−0.79	1	1	0.917	0.964	1	0.836	3122	6652
Blood coagulation	35	35	0.24	−0.19	0.82	−1.1	1	0.94	0.81	0.4	1	0.833	6444	6951
Coagulation	35	35	0.24	−0.19	0.82	−0.98	1	0.991	0.79	0.75	1	0.684	6444	6951
Cell division	18	18	0.19	−0.33	0.59	−1.3	1	0.548	0.968	0.158	0.998	0.326	4143	6233
DNA packaging	23	23	0.3	−0.21	0.97	−0.98	1	0.991	0.532	0.577	1	0.782	5378	7345
Cytokinesis	16	16	0.2	−0.34	0.6	−1.46	1	0.229	0.964	0.02	1	0.159	448	6088
ER Golgi intermediate compartment	16	16	0.35	−0.22	1.06	−0.94	1	0.994	0.39	0.566	0.923	0.712	5924	7226
Neuropeptide binding	17	17	0.29	−0.2	0.89	−0.93	1	0.994	0.67	0.679	1	0.648	4808	6250
Neuropeptide receptor Activity	17	17	0.29	−0.2	0.89	−0.77	1	1	0.638	0.853	1	0.792	4808	6250
Peptide receptor activity	38	38	0.29	−0.15	1.01	−1	1	0.988	0.477	0.5	0.99	0.84	4920	6917
Rhodopsin like receptor activity	80	80	0.23	−0.15	0.86		1	0	0.834		1	1	5403	7907

Gene # is the number of genes in the pool. ES is the enrichment score. NES is the normalized enrichment score. FWER*p* is the Family Wise Error Rate *p*-value. N*p* is the Normalized *p*-value. FDR *q* is the False Discovery Rate *q*-value. Rank is the Rank at Maximum.

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
