# Peer review of "Effects of Space Flight on Mouse Liver versus Kidney: Gene Pathway Analyses"

_ijms, 2018, doi:10.3390/ijms19124106_

Round 1
Reviewer 1 Report
Review: Hammond et al.: Effects of space flight on mouse liver vs kidney: Gene 2 pathway analyses
Hammond et al. provide very interesting data about effects on transcriptome on two different organs (kidney and liver) of mice after space flight. Most available data in which gene arrays and GO-enrichment studies were employed were achieved with cell cultures. Strength of this MS is the use of organs of space-flown mice, which is much more relevant for conclusion regarding human and animal physiology. The data obtained are in accordance with results from other researchers (e.g. effects on immune system or similar groups of altered genes compared to other objects tested so far). Very interesting is the comparison between the two organs, which share similarities in gene expression changes in some areas. The MS and the supplemental information is from high interest for the field of gravitational research and will enable further data mining in order to find common patterns in living organism induced by space-flight and microgravity.
I strongly recommend publication of the MS and have only some minor remarks, which the authors may, kindly address:
Although “space mice” and “ground-control mice” were kept in same containments, microgravity surely influenced the animals on various levels. Next to the bare typical µg-effects, which also act on cell cultures, mice in µg probably were exposed to severe stress, which also may cause (e.g .stress hormones, reduced food uptake etc.) some of the observed transcription changes. Maybe the authors should critically discuss these issues and (if data rea available) compare their transcription profile with those of stress mice under 1 g.
Please verify the reference style and font size.
Author Response
I strongly recommend publication of the MS and have only some minor remarks, which the authors may, kindly address:
Although “space mice” and “ground-control mice” were kept in same containments, microgravity surely influenced the animals on various levels. Next to the bare typical µg-effects, which also act on cell cultures, mice in µg probably were exposed to severe stress, which also may cause (e.g. stress hormones, reduced food uptake etc.) some of the observed transcription changes. Maybe the authors should critically discuss these issues and (if data are available) compare their transcription profile with those of stress mice under 1 g.
We are not aware of available radiation studies without big differences in species, sex, age, and type and duration of radiation exposure. We have added the following text on page 11; Lines 90-93: “ Space flight is a complex mixture of changes in gravity, convection, radiation exposure and other stressors. While we do not have detailed similar date on the isolated stimuli for comparison, changes in cell cycle, DNA packing, and oxygenation are consistent with changes in these pathways observed in a number of other systems [14].”
Please verify the reference style and font size.
Corrected and verified
Thank you for the suggestions, which have been incorporated wherever practical.
Reviewer 2 Report
This manuscript reports gene expression change in kidney and liver harvested from mice flown in space compare to ground control animals. Authors found common gene expression pathways activated in both Kidney and liver, and also found tissue specific pathways. The topic is appropriate for the journal and this special issue.
This is a nice piece of work and well presented, which adds new data to this research field, especially this it the first report with the kidney. I have no major comments and support the manuscript upon consideration of below
1. Is there individual variability in 8 mice from each group? It is not clear whether or how results of 8 animals in each group were pooled.
2. Is there any information with differences between different strains (strain susceptibility)?
3. The kidney results were compared with space flight results as well as simulated microgravity experimental results. Any comparison with space radiation induced gene expressions? Space flight is the combination of diverse stimuli, it is suggested to discuss about comparison with other possible contribution, like space radiation.
Author Response
This is a nice piece of work and well presented, which adds new data to this research field, especially this it the first report with the kidney. I have no major comments and support the manuscript upon consideration of below
1. Is there individual variability in 8 mice from each group? It is not clear whether or how results of 8 animals in each group were pooled.
The methods currently read on page 2 lines 54 – 58: “GSEA provides a computational method to determine whether an a priori defined set of genes shows statistically significant, concordant differences between space flight and ground controls. GSEA calculates a score for the enrichment of an entire set of genes rather than single genes, does not require setting a cutoff, allowing analysis of the complete data set, identifies the set of relevant genes as part of the analysis, and provides a robust statistical framework”
For pooling clarity we have added on page 2, lines 58 and 59: “Using modified Fisher’s exact test the replicates are integrated into the analysis so that variance between mice is accounted for no data is lost.”
2. Is there any information with differences between different strains (strain susceptibility)?
As we are sure you are aware NASA is asking for strain, sex, and age comparisons in current NRA calls. We currently have no such data available to us to make these important strain analyses.
3. The kidney results were compared with space flight results as well as simulated microgravity experimental results. Any comparison with space radiation induced gene expressions? Space flight is the combination of diverse stimuli, it is suggested to discuss about comparison with other possible contribution, like space radiation.
We are not aware of available radiation studies without big differences in species, sex, age, and type and duration of radiation exposure. We have added the following text on page 11; Lines 90-93: “ Space flight is a complex mixture of changes in gravity, convection, radiation exposure and other stressors. While we do not have detailed similar data on the isolated stimuli for comparison, changes in cell cycle, DNA packing, and oxygenation are consistent with changes in these pathways observed in a number of other systems [14].”
Thank you for these important suggestions.